# Impact of Follicle Stimulating Hormone Receptor (FSHR) Polymorphism on the Efficiency of Co-Treatment with Growth Hormone in a Group of Infertile Women from Romania

**DOI:** 10.3390/diagnostics12102371

**Published:** 2022-09-29

**Authors:** Mihai Surcel, Bogdan Doroftei, Iulia Adina Neamtiu, Daniel Muresan, Gabriela Caracostea, Iulian Goidescu, Adelina Staicu, Georgiana Nemeti, Michael S. Bloom, Cristina Zlatescu-Marton

**Affiliations:** 11st Department of Obstetrics and Gynecology, “Iuliu Hatieganu” University of Medicine and Pharmacy, 3-5 Clinicilor Street, 400006 Cluj-Napoca, Romania; 2Faculty of Medicine, “Grigore T. Popa” University of Medicine and Pharmacy, 16 University Street, 700115 Iasi, Romania; 3Health Department, Environmental Health Center, 58 Busuiocului Street, 400240 Cluj-Napoca, Romania; 4Faculty of Environmental Science and Engineering, Babes-Bolyai University, 30 Fantanele Street, 400294 Cluj-Napoca, Romania; 5Department of Global and Community Health, George Mason University, 4400 University Drive, MS 5B7, Fairfax, VA 22030, USA; 6“Regina Maria” Hospital, 29 Dorobantilor Street, 400117 Cluj-Napoca, Romania

**Keywords:** growth hormone, infertility disorder, in vitro fertilization, polymorphism, Romania, women

## Abstract

“Poor responders” (PR) are an important category of infertile women who experience a modest response to controlled ovarian stimulation. In this study, we evaluated response to growth hormone (GH) administration among PR patient subtypes stratified by follicle stimulation hormone receptor (FSHR) polymorphism (c.2039A > G p.Asn680Ser). We conducted a cohort study of 125 women with poor ovarian response, 58 of whom received GH in addition to the standard treatment, and 67 of whom received the standard treatment only. The Ala307Thr polymorphism genotypes were analyzed using a polymerase chain reaction-restriction fragment length polymorphism method, and the FSHR gene polymorphism was analyzed using a predesigned TaqMan SNP Genotyping Assay (rs6166). A comparative analysis detected statistically significant differences in mean mature follicles (*p* = 0.0002), metaphase-II oocytes (*p* = 0.0005), progesterone levels (*p* = 0.0036), and IGF levels (follicle IGF1, *p* = 0.0004) between GH-treated and non-GH-treated participants with the FSHR (Ser/Ser) polymorphism. However, the differences were modest among participants with the other two FSHR polymorphisms (Ser/Asn and Asn/Asn). The subcategory of patients with the FSHR Asn680Ser (Ser/Ser) polymorphism showed a stronger response when GH was added to the IVF protocol.

## 1. Introduction

Patients diagnosed as “poor responders” (PR) are an important category of infertile women primarily characterized by a modest response to controlled ovarian stimulation. Despite a large number of studies on this topic and multiple proposed approaches for treatment, successful IVF outcomes are infrequent, which is frustrating for both patients and physicians [1,2,3,4,5]. A likely explanation for this difficulty is the polymorphic character of infertility disorder. There are many factors that impact ovarian response to controlled stimulation, including primary oocyte anomalies, disrupted mitosis, problems with cellular differentiation or atresia, and excessive production of reactive oxygen species, as well as dysfunctions in the process of hormone-related folliculogenesis, hormone receptor function, and intercellular communication [1,6,7].

The Follicle Stimulating Hormone Receptor (FSHR) system is of particular interest to PR due to its role in the development and maturation of follicles. The FSHR belongs to the G-protein family and is expressed by granulosa cells. Receptor activation is followed by a cascade of biochemical processes involving a large number of proteins that activate genes or are otherwise involved in intracellular processes. Activated genes regulate cell proliferation, differentiation or apoptosis, and steroidogenesis [7]. Receptor activation is also linked to other intracellular processes, such as the stimulation of vascular endothelium growth factor (VEGF) [8], hypoxia-inducible factor 1 (HIF 1) [9], and insulin-like growth factor 2 (IGF 2), and the production of inhibin A [10].

The functionality of the FSHR is complex, due both to its intimate connections with other hormones such as luteinizing hormone (LH), growth hormone (GH), androgens, and insulin-like growth factor 1 (IGF 1) and its intercellular communication network [10,11]. There is a classic FSHR G-protein alpha subunit/cyclic adenosine monophosphate/protein kinase A (Gsa/cAMP/PKA) signaling pathway. However, more recently, a series of accessory pathways have been described, mediated by G-protein subtypes Gh, Gi, and Gq/11 and receptor-associated proteins (b-arrestins) [12]. FSH exposure duration and intensity, as well as other associated factors (e.g., LH, androgens, and GH levels), may substantially influence the activation or inhibition of certain pathways and the consequent biologic expression of the cell [12,13]. Thus, the study of FSHR mutations and polymorphisms represents a potentially important area of research focus, especially for patients with suboptimal ovarian response.

Hundreds of common variants or single-nucleotide polymorphisms (SNPs) of FSHR have been described. Of these, the FSHR Asn680Ser polymorphism is most frequently encountered and the most extensively studied [14,15,16,17]. In fact, there are two variants of this polymorphism in linkage disequilibrium, one located in the extracellular domain, at position 307, occupied by either alanine (Ala) or threonine (Thr), and the other located in the intracellular domain, at position 680, occupied either by asparagine (Asn) or serine (Ser). Study results regarding the influence of this polymorphism on in vitro fertilization (IVF) outcomes have been contradictory. Some authors have reported greater basal FSH levels, reduced FSH dose requirements, and, in certain cases, higher pregnancy rates in patients homozygous for serine at 680 [18,19,20]. These findings have not been confirmed by other authors [21,22]. Heterogeneous study populations and the coexistence of other dysfunctions in the systems involved in folliculogenesis are most frequently cited as reasons for the discordant results of studies [23,24].

Among numerous potential adjuvant therapies, GH treatment is one of few to have a documented benefit in terms of improved clinical pregnancy and live birth rates [25,26,27,28,29]. Even if the biological mechanism driving the reported effect has not been documented, reasonable argumentation favors IGF system involvement. GH treatment leads to an immediate increase in IGF-1, consecutive follicular development, and greater local androgen levels [30]. GH has also been shown to improve oocyte mitochondrial function in patients older than 35 years of age [31]. However, reported results from studies of GH treatment have been somewhat disappointing, and any benefit obtained must be weighed against the increased financial cost of the procedure. However, the inconsistent results from studies of GH treatment might be explained in part by a mixed patient population of PR subtypes, some of whom did not respond to GH, thereby diluting the reported effect.

The aim of this study was to evaluate the response to GH administration of different PR patient subtypes defined by FSHR polymorphisms (c.2039A > G p.Asn680Ser). Furthermore, we examined the members of the IGF family to shed light on one of the assumed action pathways of GH treatment.

## 2. Materials and Methods

### 2.1. Study Design and Population Sample

We conducted a prospective cohort study within the Assisted Reproduction Department of the 1st Obstetrics and Gynecology Clinic (Cluj-Napoca, Romania) between May 2016 and June 2021. The study was conducted according to the guidelines of the Declaration of Helsinki and approved by the Ethics Committee of “Iuliu Hatieganu” University of Medicine and Pharmacy (protocol no. 222/10 May 2016).

The study included women with poor ovarian response who were referred to our unit for IVF treatment during the indicated time frame. PR was defined according to the Bologna consensus criteria, requiring two of the following: (i) advanced maternal age (>40 years) or other risk factors for poor ovarian response; (ii) a previous poor ovarian response (<3 oocytes retrieved following a conventional stimulation protocol); and/or (iii) an abnormal ovarian reserve test (antral follicle count < 7 or anti-Müllerian hormone (AMH) < 1.1 ng/mL).

Exclusion criteria included: (i) basal FSH level greater than 15 IU/L; (ii) diagnosed with systemic lupus erythematosus, hyper/hypothyroidism, hyperprolactinemia, or having uncontrolled diabetes mellitus; and (iii) being treated with androgens or LH or supplementation with antioxidants, such as coenzyme Q10.

Women were informed about the possible benefits of GH administration and the study protocol before enrollment. Informed consent was obtained from all patients participating in the study, prior to their participation. A total of 125 women were enrolled in the study and divided into two groups: group A (58 patients), in whom GH treatment was added to the conventional treatment, and group B (67 patients), who refused the addition of GH to the conventional treatment (Figure 1). Data on the medical history of each participant were collected at the time of enrollment into the study.

### 2.2. Ovarian Stimulation Protocols, Oocyte Retrieval and Embryo Transfer

Gonadotropin (recombinant FSH) starting doses were adjusted according to patient age, body mass index (BMI), antimullerian hormone (AMH) level, and previous experience with IVF procedures. Patients received AgGnRh (Triptoreline, 0.1 mg) for down regulation in a long protocol, or Cetrorelix (Cetrotide, 0.25 mg or Orgalutran, 0.25 mg) in an antagonist protocol. In addition to the conventional treatment, group A received Somatropin (GH), 4 mg/day, subcutaneously, from the second day of the IVF cycle until administration of human chorionic gonadotrophin (hCG), while group B only received the conventional treatment. Follicular growth was evaluated by ultrasound examination on days 6, 8, 10 and sometimes, on day 12 of the IVF cycle. If no follicles ≥12 mm in diameter were identified after 10 days of gonadotrophin administration, the cycle was cancelled. When three ovarian follicles ≥17 mm were identified, hCG (Ovitrelle, 250 μg) was administered to enhance final oocyte maturation. Oocyte retrieval was performed 34–38 h following hCG administration. One to three embryos were transferred on day three or five following oocyte retrieval. Any remaining embryos were frozen for future use. To provide luteal phase support, progesterone was administered (Lutinus, intravaginal 100 mg, three times/day). Pregnancy was diagnosed by serum beta hCG testing 14 days after oocyte retrieval.

### 2.3. Biological Sampling and Analysis

At the time of oocyte retrieval, we collected 15 mL of blood by venipuncture (cubital vein) into vacutainers with EDTA, and 4 mL of follicular fluid from two mature follicles, 16–22 mm in diameter. The follicular fluid was centrifuged at 500× *g* for 10 min, transferred into sterile vials, and then frozen at −80 °C until analysis. Samples with blood contamination were excluded. The follicular fluid was analyzed for testosterone (T) estradiol, progesterone, VEGF, IGF 1, IGF 2, and IGFBP3 (IGF binding protein 3). Follicular levels of steroid hormones, paracrine factors and VEGF were determined using ELISA-based methods, with commercially available kits. For estradiol measurement, the EIAgen estradiol kit was used (the sensitivity of the assay was 15 pg/mL), the intra- and inter-assay coefficients of variability (CV) were 4.8% and 7.2%, respectively, and a dilution of 1:2 was performed. For testosterone measurement, the EIAgen testosterone kit was used (sensitivity of the assay was 0.2 ng/mL), the CV were 3.9% and 6.2%, respectively, and a dilution of 1:10 was performed. For progesterone measurement, the EIAgen progesterone kit was used (the sensitivity of the assay was 0.2 ng/mL), the CV were 5.8% and 7.5%, respectively, and a dilution of 1:1000 was performed. IGFI, IGFII, and IGFBP3 were determined using Diagnostic Systems Laboratories Webster TX kits. The CVs were 3.9% and 3.8% for IGFI, 3.4% and 4.1 for IGFII, and 4.1 and 4.5% for IGFBP3, respectively. VEGF was measured using ELISA using Quantikine Human VEGF Immunoassay from R&D Systems, Minneapolis, MN, USA (the sensitivity of the assay was 7 pg/mL). Intra- and inter-assay CV in samples were 4.5% and 7%, respectively, and 1:4 dilution was performed using the standard diluents provided with the kit, and run in duplicate.

The Ala307Thr polymorphism genotypes were analyzed using a polymerase chain reaction-restriction fragment length polymorphism (PCR-RFLP) method. The DNA was extracted from 10 mL of venous blood. A fragment of DNA with 307 bp, which contains rs6166, was amplified by standard PCR. We used BsrI restriction enzyme in the presence of acetylated bovine serum albumin (BSA) for digestion of the amplicons. The FSHR gene polymorphism at position 307 was analyzed using a predesigned TaqMan SNP Genotyping Assay (rs6166) (Life Technologies Corporation, Carlsbad, CA, USA) according to the manufacturer instructions. The resulting sequences were analyzed using SeqScape^®^ Software v3.0 (Applied Biosystems, Waltham, MA, USA).

### 2.4. Statistical Analysis

STATA v.17 statistical software (STATACorp LLC, College Station, TX, USA) was used to generate descriptive statistics and to compare clinical and paraclinical factors (e.g., mature follicles, metaphase II oocytes, number of good quality embryos, blood concentrations of testosterone, progesterone, and IGF), between patients who received GH treatment and the patients who did not. We tested the normality of clinical and paraclinical factors using Skewness and Kurtosis and the Shapiro–Wilk W tests, and considered those with *p* > 0.05 to be normally distributed. Then, we compared the mean values of clinical and paraclinical factors between the GH-treated and GH-untreated groups by T-test for normally distributed factors and by Wilcoxon–Mann–Whitney test to factors not distributed as normal. We performed a two-way ANOVA and a Bonferroni test to compare the clinical and paraclinical factors between FSHR polymorphism subtypes, and to test for the interaction between FSHR polymorphism and GH treatment. Statistical significance was defined as *p* < 0.05 for a two-tailed test.

## 3. Results

Table 1 presents the distribution of demographic, clinical and paraclinical factors among our study participants. There were *n* = 125 female participants, aged between 34 and 43 years, 58 (46%) of whom received GH treatment and 67 (54%) did not receive GH treatment. Twenty-three patients (18.4%) were included in the agonist stimulation protocol, while the majority (*n* = 102, (81.6%)) were included in an antagonist stimulation protocol. The minimum, maximum, median, 25 and 75 percentile values of clinical and paraclinical factors such as number of mature follicles, metaphase II oocytes, embryos formed, hormone levels (e.g., FSH, LH, estradiol, progesterone, testosterone), days of stimulation, total gonadotropin dose, endometrial thickness and IGF levels are shown in Table 1.

A comparative analysis using T tests, showed statistically significant greater mean values in the GH-treated group than in the group without GH treatment, for mature follicles, metaphase II oocytes, hormones (progesterone and testosterone) and follicle IGF1. Wilcoxon–Mann–Whitney tests also showed statistically significant greater values in the GH treated group than in the GH untreated for the number of good quality embryos (z = −2.609, *p* = 0.0091), and estradiol (z = −2.391, *p* = 0.0168) and serum IGF1 levels (z = −3.652, *p* = 0.0003).

The comparative analysis using *T* tests showed statistically significant differences between GH treated study participants with the FSHR (Ser/Ser) polymorphism compared to GH non-treated participants with the FSHR (Ser/Ser) polymorphism for mature follicles (*p* = 0.0002), metaphase II oocytes (*p* = 0.0005), fertilized oocytes (2PN) (*p* = 0.0001), progesterone levels (*p* = 0.0036) and IGF levels (for follicle IGF1, *p* = 0.0004; for IGF2, *p* = 0.0013; for IGFBP3, *p* = 0.0457) as shown in Table 2. Wilcoxon–Mann–Whitney tests showed statistically significant greater values in GH treated than GH untreated study participants with the FSHR (Ser/Ser) polymorphism for the number of good quality embryos (z = −3.374, *p* = 0.0007), estradiol (z = −2.484, *p* = 0.0130), and not significant for serum IGF1 level (z = −1.479, *p* = 0.1393).

Additionally, the comparative analysis using *T* test, showed statistically significant differences between GH treated study participants with the FSHR (Ser/Asn) polymorphism compared to GH non-treated participants with the FSHR (Ser/Asn) polymorphism for follicle testosterone levels (*p* = 0.0040) and IGF levels (IGF1, *p* = 0.0054; IGFBP3, *p* = 0.0289) (Table 2). A Wilcoxon–Mann–Whitney test displayed statistically significant greater serum IGF1 levels in GH treated than in GH untreated study participants with the FSHR (Ser/Asn) polymorphism, (z = −2.162, *p* = 0.0306).

For the study participants with the FSHR (Asn/Asn) polymorphism, the comparative analysis using T tests showed statistically significant differences between GH treated compared to non-treated participants for mature follicles (*p* = 0.0269) and testosterone levels (*p* = 0.0206) as shown in Table 2. A Wilcoxon–Mann–Whitney test displayed a statistically significant greater serum IGF1 level in the GH-treated than in the GH-untreated study participants with the FSHR (Asn/Asn) polymorphism, (z = −2.245, *p* = 0.0248).

Wilcoxon–Mann–Whitney tests showed statistically significant greater values in GH treated compared to GH untreated study participants with the FSHR (Ser/Ser) polymorphism for the number of transferable embryos (z = −2.772, *p* = 0.0056) and the fertility rate (z = −2.723, *p* = 0.0065) (Table 3).

Wilcoxon–Mann–Whitney tests did not show any differences in GH treated and GH untreated study participants for any of the investigated clinical outcomes, and for any FSHR polymorphism subtypes (Table 4).

The two-way ANOVA model results displayed in Table 5 suggest that there is a different GH treatment association with metaphase II oocytes (*p* = 0.0290), fertilized oocytes (2PN) (*p* = 0.0067), follicle IGF2 (*p* = 0.0043) and IGFBP3 (*p* = 0.0094) for at least two of the three FSHR polymorphism groups. The GH treatment appear to be significantly associated with the number of mature follicles (*p* = 0.0000), metaphase II oocytes (*p* = 0.0017), fertilized oocytes (2PN) (*p* = 0.0027), follicle progesterone (*p* = 0.0001), testosterone (*p* = 0.0000), and IGF1 (*p* = 0.0000). On the other hand, the FSHR polymorphisms appear to be significantly associated with follicle progesterone (*p* = 0.0288), testosterone (*p* = 0.0000), IGF1 (*p* = 0.0042) and IGF2 levels (*p* = 0.0042). Thus, the levels of follicle progesterone, testosterone and IGF1 appear to be significantly associated with both FSHR polymorphisms and GH treatment (Table 5). Due to its similarity with IGF1, in terms of physiological mechanism, we expected that GH would increase follicular recruitment by inducing a pseudo polycystic ovary status, following the increase in local follicular testosterone levels. Per secundam, a potential corrective action of GH on the FSHR functionality was speculated. In this respect, the factors that characterize the follicular steroidogenesis suggest the potentially direct influence of GH on the FSHR, and also an indirect influence through mediation by the IGF system (i.e., the increase in IGF2 levels—a major effector in mature follicles). Since IGF1 induces local secretion of androgens, higher levels of follicular IGF1 would lead to higher levels of follicular testosterone, and higher levels of follicular IGF2 would lead to higher levels of follicular progesterone.

In Table 6 we present the one-way analysis of variance (ANOVA) and the Bonferroni comparison test results. The Bonferroni test results suggest which FSHR polymorphism group pairs that may drive the statistically significant FSHR polymorphism effect estimate in Table 5. Thus, the Bonferroni test results showed statistically significant lower levels of follicle testosterone in FSHR (Ser/Ser) compared to the FSHR (Ser/Asn) and FSHR (Asn/Asn), as well as significantly lower levels of follicle IGF1 in FSHR (Ser/Ser) compared to FSHR (Ser/Asn) (Table 6).

## 4. Discussion

Our study results suggest the existence of a target group (a phenotypic variant of the FSHR (Ser/Ser)) in whom GH treatment was associated with improved IVF outcomes. Outcomes were very promising at the endpoints: mature follicles, metaphase II oocytes, fertilized oocytes, fertilization rate, increased embryo quality. However, for the other FSHR variants (Asn/Asn, Ser/Asn), GH appeared to have little practical utility. In the group that did not receive GH treatment, the presence of different FSHR polymorphisms did not appear to have clinical significance.

In terms of biochemical evaluation, we recorded greater serum IGF 1 levels in all patient groups treated with GH and greater follicular IGF 1 levels, especially among patients with the FSHR polymorphism (Ser/Ser). We also noted potentially accelerated steroidogenesis in parallel to the increased IGF 1 level, with statistically significant greater mean values of sex-steroid hormone levels in the GH-treated study participants with the FSHR polymorphism (Ser/Ser) subtype. Surprisingly, there was a greater IGF 2 level in the same subtype, although other IGF system members were relatively similar in the other investigated subtypes.

GH plays an important role in follicular development and maturation [30]. GH receptors are found in specialized follicular theca and granulosa cells [32], but its activity pathways are only partly understood. A number of possible mechanisms have been recognized, including a direct mechanism, supported by the presence of receptors on granulosa cells, theca cells, or the oocyte itself, as well as an indirect mechanism, mediated by IGF 1 and other components of the IGF system [30], and likely by the FSHR, the LH receptor or the bone morphogenetic protein receptor 1B (BMPR1B) as well [33].

The GH receptor is activated by the Janus kinase-signal transducer and activator of transcription (JAK-STAT) system, followed by tyrosyl phosphorylation and the activation of a series of transcriptional factors such as Stats 1, 3, 5a, 5b; insulin receptor substrate (IRS); extracellular signal related kinase (ERK); or mitogen-activated protein kinase (MAPK). These factors are subsequently responsible for the activation of the Ras, c-Fos, IGF 1, IRS 1, and IRS 2 genes that are involved in mitosis and apoptosis [33,34]. A number of these cellular systems are also used by the FSHR (MAPK, ERK) [12,13].

Considering these FSH-GH interconnections, we speculate that in certain PR patient subcategories, in whom at least part of the suboptimal ovarian response is attributed to FSHR dysfunction, the addition of GH to conventional treatment may reactivate certain intracellular communication pathways. Our study results support this assumption by showing a potential association between better follicular performance and GH administration in patients with the FSHR (Ser/Ser) polymorphism. This hypothesis is further supported by our results in which the other FSHR variants (Ser/Asn and Asn/Asn) (more sensitive to FSH) had only a modest response to GH administration.

The presence of FSHR polymorphisms, which render FSH administration less effective at the level of specialized ovarian cells, inevitably leads to a decrease in their activity. While inpatients with normal ovarian reserve this deficit could compensated by increasing FSH doses, in patients with an altered ovarian reserve, the increased basal levels of FSH do not allow sufficient space for cell function “optimization”. In this context, GH may offer an alternative pathway to access the follicular theca and granulosa cells and stimulate IGF1 production and consequently androgen production (as estrogen precursors). Multiple studies have identified mechanisms through which GH can improve IVF outcomes, such as enhanced ovarian reactivity by the activation of the IGF system [30], better oocyte quality by reducing oxidative stress [31], improved GH receptor expression on granulosa cells, and increased endometrial quality [35]. Results have shown associations between GH treatment and higher follicular recruitment rates, increased numbers of oocytes retrieved, better embryo quality, and improved pregnancy rates [31,33,35]. However, concern persists regarding failures to confirm clinical results or proposed action pathways and contradictory results, with some authors reporting only increased follicle recruitment and others showing only improved oocyte quality.

Taking into account the diverse phenotypic nature of PR patients, GH administration might be recommended for PR subgroups characterized by a decrease in specialized follicle cell function, but may be of little use or even detrimental in patients with very active ovarian follicles. From the clinician’s point of view, our study suggests that there may be a PR patient subcategory that may benefit from GH administration in addition to conventional treatment. Additionally, this study suggests that there could be a distinct pathogenic entity associating a functional FSHR polymorphism and the PR phenotype, reflected by different associations between clinical and paraclinical factors and GH treatment among women with different FSHR-Ala307Thr polymorphisms. However, there is still a dilemma regarding the presence of several other biologically unfavorable genetic variants (polymorphisms) that do not have clear clinical equivalents, possibly on account of compensatory mechanisms. The association of these anomalies with other pathologic elements may, however, enter the territory of clinical conditions. In this respect, identifying different polymorphisms inside well-defined clinical syndromes, together with testing responses to different therapies, will allow for a better understanding of the pathophysiology of PR and help clinicians refine treatment for its management. Tailoring the optimal dose of gonadotropins seems to be very important, particularly in patients with ovarian dysfunctions such as PR. Over the past 30 years, to help practitioners individualize treatment, the use of several predictors of ovarian reserve/response to controlled stimulation, either individually (e.g., day 3 FSH, AMH, inhibin B, AFC) or in combinations, such as the CONSORT formula or ovarian response prediction index (ORPI; AMH level x AFC / patient’s age), have been proposed [36,37,38,39,40]. Several studies have reported that AMH, AFC, and ORPI are effective predictors of poor ovarian response, and that ORPI was the most effective predictor of ovarian hyperstimulation syndrome [36,37]. However, these predictors are limited by the complexity of folliculogenesis, in which genetically or epigenetically induced intra- and intercellular dysfunction (e.g., different polymorphisms) may interfere with the normal process of follicular development. As such, patients with complex dysfunction may fall outside of the responses to controlled ovarian stimulation predicted by these factors.

Our study has several limitations. Major limitations include the low sample size, no randomizations, and the fact that we used multiple hypothesis tests and did not adjust for confounding. Another limitation is that we only evaluated some of IGF system proteins, while several other proteins involved in the regulation of this system, such as IGFBP 2,4,5,6 and the respective proteases, were not part of this evaluation. Despite their limitations, we consider that our study results may help lay the groundwork for future research in this area (the use of GH in poor responders with Ser/Ser variant of FSHR 680 polymorphism), supporting the use of GH treatment in clinical practice. Our study results suggest that there was an association with folliculogenesis, reflected by both clinical (number of mature follicles, number of retrieved oocytes, number of good quality embryos), and paraclinical factors (higher levels of IGF2, which plays a major role in the final stages of folliculogenesis). Additionally, we noted a greater pregnancy rate (30% vs. 11.8%), although this was not statistically significant. This was not unexpected, considering the small sample size in each group of patients with the three FSHR polymorphisms. Even in available meta-analyses, which were based on randomized trials, due to the small sample size of the studies included, the reported results (which support the use of GH therapy, as it was associated with increased pregnancy and live birth rates) are not statistically significant [26,41]. For other clinical factors such as implantation, abortion, or live birth rate, our results did not show associations with GH therapy. This was anticipated to some extent, since important factors (e.g., adenomyosis, junctional zone thickness, endometrium markers of chronic endometritis) were not addressed in this study. Our results are consistent with those reported by experts in this field (members of the ESHRE board who elaborated the most recent ESHRE guidelines for best clinical practice in ovarian stimulation) who recommend the use of GH therapy in certain groups of patients, based on small sample size studies, where not all the major outcomes turned out to be statistically significant (live birth rate, in particular) [42,43].

## 5. Conclusions

Adding GH to the IVF protocols may be justified in certain PR patient subgroups, and women with the FSHR Asn680Ser (Ser/Ser) polymorphism subtype may be “candidates” for this treatment, but more extensive and comprehensive studies, including randomized controlled trials, are necessary to confirm these results. Additionally, studies including the analysis of genetic polymorphisms related to other receptors and sex hormones within the GH or FSH action pathways may help to identify additional “candidates” for this therapy.

## Data Availability

Not applicable.

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
