# Peer review of "Impact of Follicle Stimulating Hormone Receptor (FSHR) Polymorphism on the Efficiency of Co-Treatment with Growth Hormone in a Group of Infertile Women from Romania"

_diagnostics, 2022, doi:10.3390/diagnostics12102371_

Round 1

Reviewer 1 Report

In my opinion, there are not basic IVF data for supporting the main hypothesis of this study. Please consider my suggestions.

1.     Please show the comparisons of patient characteristics of each group and consider followings. It will be better to merge of Table 1, 2 and 3 into New Table 1.

: FSHR subtypes of Ser/Ser vs Asn/Asn vs Ser/Asn in GH group vs. Control group.

Age, Infertility year, BMI, FSH, LH, E2, Progesterone,

Days of gonadotropins, total gonadotropin dose

Endometrial thickness, stimulation protocol type

2.     In accordance with New Table 1, please show the detailed results of the IVF results in Table 4, 5 and 6, which merged into New Table 2 including the followings.

:Number of 2PN oocytes, fertilization rate, cleavage rate,

Number of transferable embryos, transferable embryo rate,

Number of transferred embryo per ET, Cycle reaching ET rate

Failed fertilization number, failed cleavage, OHSS

3.     Show the implantation rates, clinical pregnancy rate, Early miscarriage rate, Fresh or Frozen ET rate, Ongoing pregnancy rate and Livebirth rate in New Table 3.

Author Response

Our responses to the questions raised by the Reviewer are in the attached file. Please see the attachment.

Reviewer 2 Report

I read with great interest the Manuscript titled "Impact of Follicle Stimulating Hormone Receptor (FSHR) polymorphism on the efficiency of co-treatment with growth hormone in a group of infertile women from Romania" which falls within the aim of the Journal.
In my honest opinion, the topic is interesting enough to attract the readers’ attention. Methodology is accurate, conclusions are supported by the data analysis, the discussions are well structured and the tables are clear and understandable.

Overall the work is well organized, the language used is appropriate.

For all the reasons listed above I am glad to accept this job in the present form.

Author Response

Response to Reviewer

Enclosed: Manuscript Ref. No.: diagnostics-1876421

Impact of Follicle Stimulating Hormone Receptor (FSHR) polymorphism on the efficiency of co-treatment with growth hormone in a group of infertile women from Romania

Mihai Surcel, Bogdan Doroftei, Iulia Neamtiu, Daniel Muresan, Gabriela Caracostea, Iulian Goidescu, Adelina Staicu, Georgiana Nemeti, Michael S. Bloom, Cristina Zlatescu-Marton

Comments and Suggestions for Authors

I read with great interest the Manuscript titled "Impact of Follicle Stimulating Hormone Receptor (FSHR) polymorphism on the efficiency of co-treatment with growth hormone in a group of infertile women from Romania" which falls within the aim of the Journal.
In my honest opinion, the topic is interesting enough to attract the readers’ attention. Methodology is accurate, conclusions are supported by the data analysis, the discussions are well structured and the tables are clear and understandable.Overall the work is well organized, the language used is appropriate.

For all the reasons listed above I am glad to accept this job in the present form.

Please find our responses below, in Italics, labeled and bolded as “Response”.

 Response: We would like to thank the Reviewer for the positive feedback on our manuscript. We revised our Reference section and added more relevant references in our resubmitted manuscript (pages 16-18). Also, before resubmission, our manuscript was critically revised and proofread by a native English-speaking colleague at a U.S. academic and research institution.

Due to multiple English language changes made by our English-speaking colleague, our manuscript text has become difficult to edit and read, so we asked for and received the permission from the Assistant Editor, Ms. Georgiana Ilies, to “accept” the language changes and only keep in “track” the changes made in response to the Reviewer’s specific comments and suggestions. If the Reviewer requests, we can provide a copy of the manuscript with all English language changes in “track”.

Reviewer 3 Report

I read with great interest the manuscript, which falls within the aim of this Journal. In my honest opinion, the topic is interesting enough to attract the readers’ attention. Nevertheless, authors should clarify some points and improve the discussion, as suggested below.

Authors should consider the following recommendations:

-       Manuscript should be further revised in order to correct some typos and improve style.

-        I recommend to add further discussion about the use of age, FSH, AMH, antral follicle count (AFC), and ovarian response prediction index (ORPI), as potential predictors of response to controlled ovarian stimulation (authors may refer to: PMID: 32613875; PMID: 30242498).

Author Response

Response to Reviewer

Enclosed - Manuscript: Ref. No.: diagnostics-1876421

Impact of Follicle Stimulating Hormone Receptor (FSHR) polymorphism on the efficiency of co-treatment with growth hormone in a group of infertile women from Romania

Mihai Surcel, Bogdan Doroftei, Iulia Neamtiu, Daniel Muresan, Gabriela Caracostea, Iulian Goidescu, Adelina Staicu, Georgiana Nemeti, Michael S. Bloom, Cristina Zlatescu-Marton

Thank you for the suggestions and constructive comments. The critiques you provided have helped us to clarify and improve our work. Please find our responses in detail below, for all the questions raised by the Reviewer, in Italics, labeled and bolded as “Response”. Revisions made in response to the reviewer’s comments and suggestions are marked as track changes in our resubmitted manuscript.

Comments and Suggestions for Authors

I read with great interest the manuscript, which falls within the aim of this Journal. In my honest opinion, the topic is interesting enough to attract the readers’ attention. Nevertheless, authors should clarify some points and improve the discussion, as suggested below.

Authors should consider the following recommendations:

Point 1: Manuscript should be further revised in order to correct some typos and improve style.     

Response: We would like to thank the Reviewer for the positive feedback on our manuscript and for providing us with further suggestions and guidance. As the Reviewer suggested, before resubmission, our manuscript was critically revised and proofread by a native English-speaking colleague at a U.S. academic and research institution.

Due to multiple English language changes made by our English-speaking colleague, our manuscript text has become difficult to edit and read, so we asked for and received the permission from the Assistant Editor, Ms. Georgiana Ilies, to “accept” the language changes and only keep in “track” the changes made in response to the Reviewer’s specific comments and suggestions. If the Reviewer requests, we can provide a copy of the manuscript with all English language changes in “track”.

Point 2: I recommend to add further discussion about the use of age, FSH, AMH, antral follicle count (AFC), and ovarian response prediction index (ORPI), as potential predictors of response to controlled ovarian stimulation (authors may refer to: PMID: 32613875; PMID: 30242498).

Response: We would like to thank the Reviewer for the positive feedback on our manuscript and for providing us with further suggestions and guidance.

As the Reviewer suggested, we also added further discussion on the use of age, FSH, AMH, antral follicle count (AFC), and ovarian response prediction index (ORPI) as potential predictors of response to controlled ovarian stimulation and referred to the articles indicated. Please see below and in our revised manuscript (page 15, lines 524-536). All references in the new paragraph added to the Discussion section were also included in the References section, in our resubmitted manuscript (page 18).

“Tailoring the optimal dose of gonadotropins seems to be very important, particularly in patients with ovarian dysfunctions such as PR. Over the past 30 years, to help practitioners individualize treatment, the use of several predictors of ovarian reserve/response to controlled stimulation, either individually (e.g., day 3 FSH, AMH, inhibin B, AFC), or in combinations, such as the CONSORT formula or ovarian response prediction index (ORPI; AMH level x AFC / patient’s age), have been proposed [1-5]. Several studies reported that AMH, AFC and ORPI were effective predictors of poor ovarian response, and that ORPI was the most effective predictor of ovarian hyperstimulation syndrome [1,2]. However, these predictors are limited by the complexity of folliculogenesis, in which genetically or epigenetically induced intra- and intercellular dysfunction (e.g., different polymorphisms) may interfere with the normal process of follicular development. As such, patients with complex dysfunction may fall outside of the responses to controlled ovarian stimulation predicted by these factors”.

References citied

  1. Ng, D.Y.T.; Ko, J.K.Y.; Li, H.W.R.; Lau, E.Y.L.; Yeung, W.S.B.; Ho, P.C.; Ng, E.H.Y. Performance of Ovarian Response Prediction Index (ORPI) in Predicting Ovarian Response and Livebirth in the in-Vitro Fertilisation Cycle Using a Standard Stimulation with Corifollitropin Alpha in a GnRH Antagonist Protocol. Human Fertility 2022, 25, 377–383, doi:10.1080/14647273.2020.1805517.
  2. Ashrafi, M.; Hemat, M.; Arabipoor, A.; Salman Yazdi, R.; Bahman-Abadi, A.; Cheraghi, R. Predictive Values of Anti-Müllerian Hormone, Antral Follicle Count and Ovarian Response Prediction Index (ORPI) for Assisted Reproductive Technology Outcomes. Journal of Obstetrics and Gynaecology 2017, 37, 82–88, doi:10.1080/01443615.2016.1225025.
  3. Peluso, C.; Oliveira, R. de; Laporta, G.Z.; Christofolini, D.M.; Fonseca, F.L.A.; Laganà, A.S.; Barbosa, C.P.; Bianco, B. Are Ovarian Reserve Tests Reliable in Predicting Ovarian Response? Results from a Prospective, Cross-Sectional, Single-Center Analysis. Gynecological Endocrinology 2021, 37, 358–366, doi:10.1080/09513590.2020.1786509.
  4. di Paola, R.; Garzon, S.; Giuliani, S.; Laganà, A.S.; Noventa, M.; Parissone, F.; Zorzi, C.; Raffaelli, R.; Ghezzi, F.; Franchi, M.; et al. Are We Choosing the Correct FSH Starting Dose during Controlled Ovarian Stimulation for Intrauterine Insemination Cycles? Potential Application of a Nomogram Based on Woman’s Age and Markers of Ovarian Reserve. Archives of Gynecology and Obstetrics 2018, 298, 1029–1035, doi:10.1007/s00404-018-4906-2.
  5. Lensen, S.F.; Wilkinson, J.; Leijdekkers, J.A.; la Marca, A.; Mol, B.W.J.; Marjoribanks, J.; Torrance, H.; Broekmans, F.J. Individualised Gonadotropin Dose Selection Using Markers of Ovarian Reserve for Women Undergoing in Vitro Fertilisation plus Intracytoplasmic Sperm Injection (IVF/ICSI). Cochrane Database of Systematic Reviews 2018, 2, CD012693, doi:10.1002/14651858.CD012693.pub2.

Round 2

Reviewer 1 Report

Dear authors

1. I recognize your idea is very interesting. Nevertheless, your way of description in tables is not intuitive and straightforward. All tables deal with critical information about patients, but the tabular organization you designed impedes clear understanding of what you say. Please make sure between-comparison for each index (i.e., comparison between groups [Ser/Ser, Asn/Asn, Ser/Asn and Control]. Here is a good example of the presentation [Please see tables in https://doi.org/10.1186/s12884-020-03004-9]. 

2. The authors should describe what is the unit of the values (e.g., the number of eggs or concentrations or anything else?) in all tables for the reader’s clear understanding. I know that table 1 already had such descriptions except for several items. However, table 2 and others had such no description. Please specify the unit of each item for clearness as possible.

3. All statistical comparison were conducted within groups (e.g., YES or NO of each subtype). Based on the observations, the authors made a conclusion. What about between-group comparison? Are you sure whether a statistical significance calculated from within-group comparison can make it clear to conclude your hypothesis? I recommend you perform statistical comparison among Ser/Ser, Ser/Asn, Asn/Asn and Control.

4. The author claimed that Ser/Ser subtype is superior to other groups in terms of GH-dependent fertility. To support the author’s idea, they showed some of rates in tables 4 and 5. What about the absolute values (or counts)? For example, how can you calculate pregnancy rate from what absolute cases of what total? Please specify the absolute value with the normalized rate in the tables.

In addition, I did not point out English but a way of writing (mostly from tabular descriptions). Please consider structural writing of the manuscript. If possible, specific description of your points by visualization using a boxplot or a histogram can be one of solutions for clear and concise delineation.

Round 3

Reviewer 1 Report

Dear Authors

I think a big improvement in the revised manuscript. I appreciate the authors’ efforts. In overall, the manuscript has been relatively well revised than before. However, there are required some amendment to accept the authors’ manuscript in the journal. In addition, I think there is a little mis-communication between authors and referees.

1.      Actually, information described in the Figure 1 of the revised manuscript (previously depicted in table 1) has to be summarized as a table. If you really have an important thing to stress among them, visualization by plotting could be reasonable to help readers to understand what is your point. The reason why I recommended you to use a plot was related to other tables.

2.      Related to the comment 1, drawing a bunch of variables with different digits (e.g., total gonadotropine dose with 4 digits and age with 2 digits) at the same panel is not encouraged because a variable with high value easily masks a possible trend of a second variable with low value. I don’t know why you transformed the features of the previous table 1 into plots, it was not my recommendation. I post an example at the bottom of the paragraph to help your preparation. 0 to 4 in Quantile mean Min, 25%, 50%, 75%, and Max, respectively. Please re-organize the previous table 1 instead of the Figure 1 in the revised manuscript.

Factor

Ser/Ser

Ser/Asn

Asn/Asn

Total

N

Quantile

Total

N

Quantile

Total

N

Quantile

0

1

2

3

4

0

1

2

3

4

0

1

2

3

4

Female ages

Y

N

3.      Tabular description is highly improved except for table 2. I am giving you an example. Placement of all groups at the same row can be well recognizable for readers.

Mean+/-SD

Subtype FSHR

GH treated

GH untreated

MD (95% CI)

P-value

Mature follicles

(Number)

(Ser/Ser)

(Ser/Asn)

(Asn/Asn)

4.      Unit description in tables was well organized except for several cases. When you describe the value as percentage, you don’t need what is the unit. If the title of the clinical outcomes (e.g., in table 5) begins with ‘No. of …’, no need to describe the unit in parentheses. 

5.      In table 4, two-way ANOVA to examine a possible contribution of two (probably under independent assumption) variables such as polymorphisms and GH treatment to each clinical index was a good approach. However, considering the primary use of two-way ANOVA, your interpretation was so simple. Particularly, you statistically mentioned an interaction between two variables. Is there a biological interaction between two factor variables upon assumption of your model hypothesis? If no statistical significancy in interaction between two inputs (e.g., follicle testosterone), is there no relevance? Please explain the result by relating the model results to biological situation.

6.      Related to the 5th comment, I’m not sure whether the statistics in table 5 were obtained from the two-way ANOVA model you’ve established for table 4. If not, you should explain contribution of each value in the variable via post-hoc method (e.g., Tukey, Dunnette or something applicable).

7.      In lines 208-214, page 6 of the revised manuscript, you wrote “A comparative analysis using T test, … (data not shown in the tables)”. Is it related to the revised table 1? Otherwise, why did you you type ‘data not shown’? Actually, I found such sentences elsewhere in the manuscript. Officially, ‘data not shown’ is not permitted in the journal. Please include the data.

8.      You conclusion is that several factors such as MII, 2PN, IGF1, etc. are affected by FSHR polymorphisms and GH treatment, and no influence on pregnancy rate, etc. The conclusion is clear. However, interpretation of their relationships (or clinical aspect) is not well discussed except for simple citation of previous reports. What is your thought of the unrelatedness between what you have found and the outcome you’ve expected?

Round 4

Reviewer 1 Report

Dear authors,

I found many positive improvement in the revised manuscript. I appreciate your efforts.

Regards.